# Thermodynamics of Hydrogen Storage: Equilibrium Study of Liquid Organic Hydrogen Carrier System 1-Methylindole/octahydro-1-methylindole



**Sergey V. Vostrikov** [1,*], **Maria E. Konnova** [1], **Vladimir V. Turovtsev** [2], **Karsten Müller** [3,4], **Jason E. Bara** [5] and **Sergey P. Verevkin** [4,6,*]

1    Chemical-Technological Department, Samara State Technical University, 443100 Samara, Russia
2    Department of Physics, Tver State Medical University, 170100 Tver, Russia
3    Institute of Technical Thermodynamics, University of Rostock, 18059 Rostock, Germany
4    Competence Centre CALOR at the Department Life, Light & Matter of the Interdisciplinary Faculty, University of Rostock, 18059 Rostock, Germany
5    Department of Chemical & Biological Engineering, University of Alabama, Tuscaloosa, AL 35487, USA
6    Department of Physical Chemistry, Kazan Federal University, 420008 Kazan, Russia
*    Correspondence: vosser@mail.ru (S.V.V.); sergey.verevkin@uni-rostock.de (S.P.V.)

**Abstract:** Methylated indoles could be potentially interesting components for hydrogen ($H_2$) storage based on the Liquid Organic Hydrogen Carrier (LOHC) approach. It is likely that the methylated forms will preserve the beneficial thermochemical characteristics of indole compared to homocyclic LOHCs. At the same time, 1-methyl-indole is expected to have a higher stability than indole in the reactive cycle of hydrogenation and dehydrogenation. This study investigates whether the expectations regarding reaction thermodynamics and stability are justified. To this end, the chemical equilibria of the hydrogenation ($+H_2$)/dehydrogenation ($-H_2$) reactions of the indole/octahydroindole LOHC system was determined experimentally over a wide range of temperature, pressure, and hydrogen:feedstock ratio. Reaction thermodynamics were calculated from the relationship between temperature and equilibrium constant. In addition, the formation enthalpies of the species involved in the reaction have been determined experimentally utilizing combustion calorimetry. Further validation has been achieved using high-level quantum chemical methods. The evaluation confirms both hypotheses: (1) 1-methyl-indole exhibits less decomposition during reaction as is the case for the indole system. Hence, an improved stability of methylated LOHC molecules can be concluded; (2) The enthalpy of reaction for $H_2$ release from octahydro-1-methyl-indole is estimated from the equilibrium experiments and calorimetric measurements to be about +55.6 kJ mol$(H_2)^{-1}$ for reaction in the liquid phase at standard conditions. This is comparable to the values observed for octahydro-indole.

**Keywords:** chemical equilibria; reaction enthalpy; reaction entropy n; QC calculations

## 1. Introduction

Liquid Organic Hydrogen Carriers (LOHCs) enable safe and dense storage of $H_2$ by reversible hydrogenation of an aromatic substance. It has been demonstrated that the LOHC system indole/octahydroindole is a promising candidate for the storage of $H_2$ [1–3], even though there are some restrictions regarding its thermal stability. However, it is known that tertiary amines are usually more stable compared to secondary amines [4]. Therefore, in this work, we extended our thermochemical studies to the LOHC system 1-methyl-indole/octahydro-1-methyl-indole. Both hydrogenated and dehydrogenated counterparts have low melting points and high stability [5,6]. Such properties of the system make it possible to successfully realize the concept of $H_2$ storage and transport.

This investigation focuses on the study of equilibrium analysis of thermodynamics of the hydrogenation ($+H_2$)/dehydrogenation ($-H_2$) reactions in the LOHC system 1-methyl-indole/octahydro-1-methyl-indole. Various partially and fully dehydrogenated species are

present in the reaction mixture (see Figure 1). Hence, knowledge of the interconversion of these intermediates and their thermodynamic stability is essential for the optimization of LOHC-based technologies.

(**a**) H0-1-methylindole (H0-MI) (1-methylindole)

(**b**) H2-1-methylindole (H2-MI) dihydro-1-methylindole (1-methylindoline)

(**c**) H4-1-methylindole (H4-MI) 4,5,6,7-tetrahydro-1-methylindole

(**d**) H8-1-methylindole (H8-MI) octahydro-1-methylindole

**Figure 1.** The intermediates identified in the equilibrium mixture of the hydrogenation in the LOHC system 1-methyl-indole/octahydro-1-methyl-indole.

## 2. Methods

### 2.1. Materials

The sample of 1-methylindole (CAS: 603-76-9) was of commercial origin (Alfa Aesar, 98+%). The liquid sample of 1-methylindoline (CAS: 88475-55-2) was prepared and purified in laboratories of the University of Alabama according to the established method described in [7]. Additional details of the synthetic methods are provided as Supporting Information. Before the combustion and transpiration experiments, the samples were further purified by fractional distillation. No impurities (mass fraction greater than 0.0003) were detected in the samples used for the thermochemical measurements. The degree of purity was determined using a GC equipped with an FID. A capillary column HP-5 with a column length of 30 m, an inner diameter of 0.32 mm, and a film thickness of 0.25 μm was used. Water mass fraction in the samples (see Table S2) was determined using a Mettler Toledo DL38 Karl Fischer titrator using the HYDRANAL™ as the reagent.

### 2.2. Chemical Equilibrium Study

The equilibria of the hydrogenation reactions in the 1-methyl-indole/octahydro-1-methyl-indole system were investigated in the range of T = 413–493 K. The experiments were performed in a 300 mL autoclave R-201 constructed from Hastelloy C-276 (origin Korea). About 2 g of 1-methylindole (substrate) was dissolved in 150 mL of n-hexane. This mixture and 6–20 g of catalyst (Pt, Pd and Ni catalyst supported on $SiO_2$) were loaded into the reactor. The reactor was sealed and purged several times with $H_2$ to remove any residual air from the system. After that, the reactor was heated to the desired temperature and brought to the desired pressure. When varying the $H_2$ pressure from 0.1 to 2.6 MPa, the ratio of $H_2$ to feed was scaled from 0 to 10. Our calculations show that for the typical volumes (50 or 150 mL) of the initial mixture in the total volume of 300 mL, the reactions actually proceed in the liquid phase at all temperature and pressure conditions.

The samples of the reaction mixture were withdrawn periodically and analyzed using the GC (Kristall-2000M, at Samara State University) equipped with FID and capillary column ZB-35 (30 m × 0.25 mm × 0.25 μm). Chemical equilibrium in the reaction system was considered to be reached when the ratios of the reactants were constant over a long period of time and were independent of the variation in the compositions of substrates and catalysts.

### 2.3. Identification of Reaction Species Present at Equilibrium

Gas chromatography–mass spectrometer (GC-MS) QP2010Ultra (Shimadzu, Japan) was used to identify products. The chromatography column had a length of 100 m and the mass detector range was 1.5 to 1090 m/Z. The sensitivities were 1 PG of octafluoroaphtha-lene S/N 500 for m/Z = 272 (SCAN mode) and 100 FG of octafluoroaphthalene S/N 500 for m/Z = 272 (SIM mode). The maximum data processing rate was 100 scans/s (i.e., 100 Hz),

whereas the maximum scanning speed was 20,000 amu/s, and the minimum scan interval was 10 ms.

Mass spectra for 1-methylindole and 1-methyl-dihydroindole are available in the literature and the data obtained are in complete agreement with them, namely: for 1-methyl-indole we recorded the spectrum 130, 103, 89, 77, 65, 51, 39, 27 and for and 1-methyl-dihydroindole 132, 117, 103, 91, 77, 65, 51, 36.

The NIST database [8] does not have a spectrum for the 1-methyl-octahydroindole. Nevertheless, the available spectrum for 3-methyloctahydroindole (*m/z*) 139, 124, 96, 82, 68, 56, 41, 30 with similar characteristics of ions was used to resolve the structure of 1-methyl-octahydroindole from the following *experimental* spectrum (*m/z*): 139, 122, 96, 82, 67, 41. Obviously, for the main characteristic ions, the data are consistent, suggesting that the identification is correct.

The following spectrum was measured for 4,5,6,7-tetrahydro-1-methylindole (*m/z*): 135, 117, 107, 92, 77, 65, 36. In the absence of spectra for comparison, it was assumed that the fragmentation of this molecule follows two mechanisms. The first mechanism is characterized by the release of a radical, which leads to the formation of the indole 117 ion. The second one is caused by the elimination of the HCN molecule, with the formation of the 107 ion.

The following spectrum was measured for 4,5,6,7-tetrahydro-1-methylindole (*m/z*): 135, 117, 107, 92, 77, 65, 36. In the absence of comparative spectra, it was assumed that the fragmentation of this molecule could follow two mechanisms. The first mechanism is characterized by the release of a radical leading to the formation of the indole-117 ion. The second is caused by the elimination of the HCN molecule to form the 107 ion. Dong et al. [9] calculated the energetics of the indole derivatives using DFT and they show that the most stable structure of 1-methyl-tetrahydroindoles is 4,5,6,7-tetrahydro-1-methylindole. The only peak found by GC-MS in the molecular weight range related to the tetrahydro-1-methylindoles. Therefore, this peak can be assumed to be 4,5,6,7-tetrahydro-1-methylindole, which confirms the data obtained by GC-MS. No signals were found in the molecular weight range of the hexahydro-1-methylindoles by means of GC-MS.

### 2.4. Experimental and Theoretical Thermochemical Methods

A high-precision static bomb calorimeter was used for measurements of standard molar combustion energies of 1-methyl-indole and 1-methyl-indoline. The detailed procedure has been described in our earlier works [10,11]. A transpiration method was used to measure the vapor pressures of 1-methylindole and 1-methylindoline. The detailed procedure was previously reported [12,13]. Measurements of heat capacity of 1-methyl-indoline were performed with a Perkin Elmer DSC Pyris 1 in the temperature range from liquid nitrogen to approximately 350 K [14]. Some essential details on the *experimental* techniques are provided in Electronic Supporting Information (ESI).

The quantum-chemical (QC) software package, Gaussian 16$^{®}$, [15] with the G4 [16], G4MP2 [17], G3MP2 [18], and CBS-APNO [19] methods, was used to calculate enthalpies ($H_{298}$) for each compound's most stable conformer. The $H_{298}$-values provided in the output files were converted to the *theoretical* gas-phase standard molar enthalpies of formation, $\Delta_f H_m^o$(g, 298.15 K)$_{theor}$, of the reactants in the 1-methyl-indole/octahydro-1-methyl-indole LOHC system and discussed. Because the G4 and G3MP2 methods are similar in composition, their potential systematic errors may be the same or similar. In addition, to independently confirm the correctness of $\Delta_f H_m^o$(g, 298.15 K)$_{theor}$ values from the Gn methods, the calculations were performed using the high-level composite method CBS-APNO. The latter method differs from Gn in a number of computational steps. Details on the calculation methods have been reported elsewhere [20]. The "rigid rotator-harmonic oscillator" approach embedded in the Gaussian 16 software was used for these calculations.

## 3. Results and Discussion

### 3.1. Equilibrium in the 1-methyl-indole/octahydro-1-methyl-indole LOHC System

The intermediates identified in the equilibrium mixture of 1-methyl-indole hydrogenation are given in Figure 2. The following stepwise hydrogenation reactions (see Figure 2) describe the reaction network for this system.

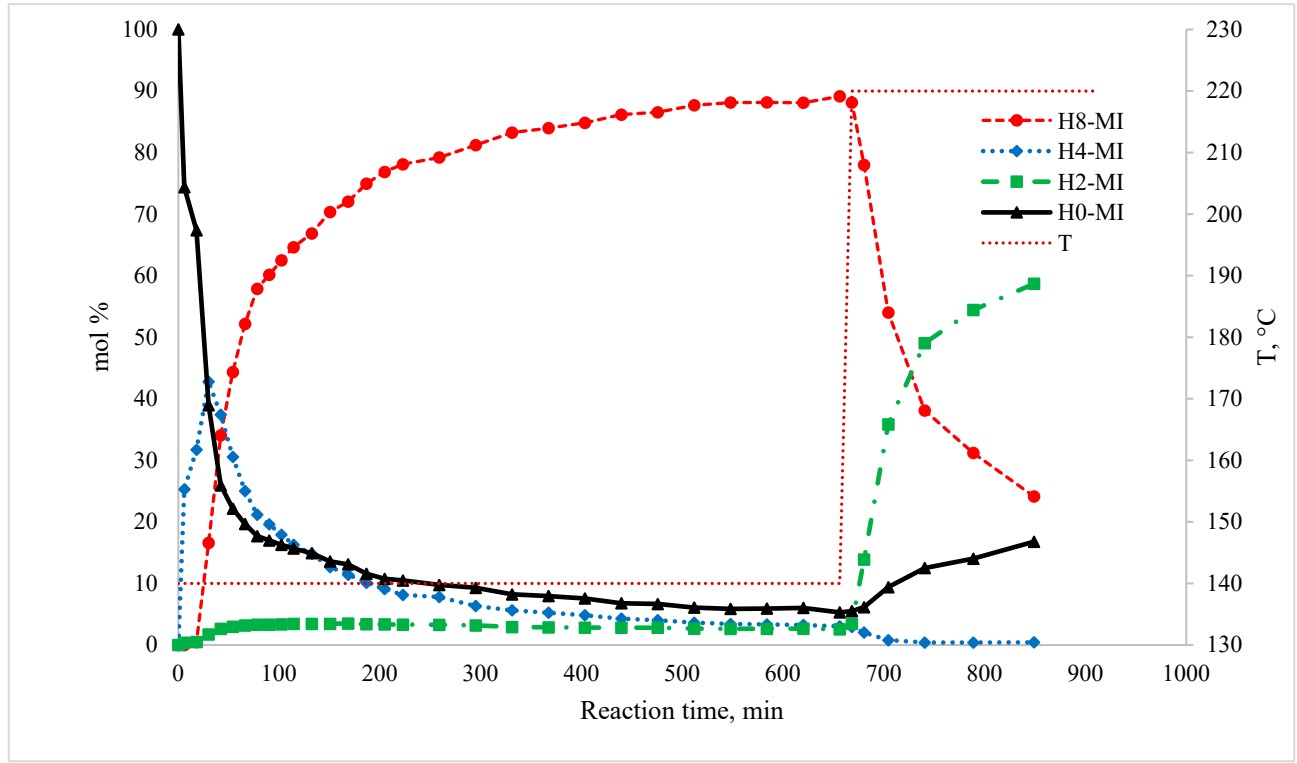

**Figure 2.** Equilibria of hydrogenation reactions for the 1-methyl-indole/octahydro-1-methyl-indole LOHC system.

Like every reaction, the hydrogenation of this LOHC system proceeds under kinetic control until the reaction approaches equilibrium. The kinetics of the hydrogenation step were not the focus of this work. Details on the kinetic behavior of this system have been reported by Stepanenko et al. [3] recently. Figure 3 illustrates the typical profile of the reaction system composition over time for the hydrogenation of 1-methyl-indole.

**Figure 3.** Plot of the typical reaction mixture composition with respect to contact time in H0-MI hydrogenation. Catalyst = 9.1% mass (12% Ni/SiO$_2$); $P_{0_{H_2}}$ = 21.3 atm; $H_2$/1-methyl-indole = 7.3 mol/mol.

The main product of the hydrogenation of 1-methyl-indole (H0-MI) at 413 K is 1-methyl-perhydroindole (H8) in an amount of up to 90 mol%. It is noteworthy that the

amount of the product of the hydrogenation of the double bond in the N-heterocycle, namely 1-methyl-indoline (H2-MI), does not exceed 1–3 mol%. Additionally, 1-methyl-hexahydroindoles (H6-MI) cannot be detected in the reaction mixture of hydrogenation of 1-methylindole. The only reaction intermediate observed is 1-methyl-4,5,6,7-tetrahydro-1H-indole (H4-MI). This product is formed in large quantities at the first moment of hydrogenation but then is further hydrogenated to the perhydro-compound H8-MI.

The primary (i.e., left) y-axis is the composition (i.e., mol% of given species, excluding solvent and $H_2$) of the reaction mixture; the secondary (i.e., right) y-axis is the temperature of the reaction in °C. The dotted line illustrates the step change in the reaction mixture temperature after equilibrium was achieved.

H0-MI: H0-1-methylindole (1-methylindole); H2-MI: H2-1-methylindole (dihydro-1-methylindole); H4-MI: H4-1-methylindole (4,5,6,7-tetrahydro-1-methylindole); H8-MI: H8-1-methylindole (octahydro-1-methylindole).

With an increase in the reaction temperature to 493 K, additional H4-MI formation does not occur; on the contrary, this product is consumed by dehydrogenation to H2-MI. Additionally, the amount of H8-MI at 493 K begins to decrease sharply. At the same time, due to the shifted equilibrium, the initial reactants H0-MI and H2-MI are formed again in significant amounts in the reaction mixture. At this temperature, the intermediate products H6-MI were not found in the reaction mixture either. Thus, the hydrogenation of H0-MI occurs exclusively via the hydrocarbon ring in the H4-MI intermediate, in contrast to the hydrogenation of indoline or indole [1], where some amount of the H6-indole intermediate was also found in the reaction mixture.

The composition of the reaction mixture found in this study agrees with the results of the DFT calculation in [9] where the analogous dehydrogenation reaction of H8-1-ethylindole was evaluated. The calculations support the conclusion that the most likely dehydrogenation pathway is via H4-1-ethylindole formation and that the pathway via H6-1-ethylindole formation is energetically less favorable. They also found only one stable H4-1-ethylindole intermediate by GC-MS, in agreement with our hydrogenation equilibrium results of the H0-MI/H8-MI system.

### 3.2. Calculation of Equilibrium Constants

After equilibrium is established, the system runs under thermodynamic control. The relative quantities of reactants and intermediates are determined by the thermochemical properties (enthalpy of formation and entropy) of the participating species. In order to ensure that the "true" state of equilibrium is reached, rationally designed experiments were performed out wherein the amounts of substrates and catalysts differed across a broad range. Chemical equilibrium can only be confirmed if the equilibrium constants remain stable and calculations are reproducible over long time periods, regardless of the catalyst type or amount, and/or the initial reaction composition.

The thermodynamic liquid phase equilibrium constants, $K_a$, from the *experimental* data are commonly derived using Equation (1) [21]:

$$K_a = \frac{a_{rp}}{a_{in} \cdot \left(\frac{f_{H_2}}{P^+}\right)^{v_{H_2}}} \tag{1}$$

where $a_{in}$ and $a_{rp}$ are the activities of the reactant/intermediate and the product, respectively; $f_{H_2}$ is the fugacity of $H_2$ and $v_{H_2}$ is the absolute value of its stoichiometric coefficient. $P^+$ is the reference pressure. The activity is equal to the product of the activity coefficient and the mole fraction of the component Equation (2):

$$a_i = \gamma_i \cdot x_i \tag{2}$$

$$K_a = K_\gamma \cdot K_x = \frac{\gamma_{rp}}{\gamma_{in} \cdot \varphi_{H_2}} \cdot \frac{x_{rp}}{x_{in} \cdot \left(\frac{P_{H_2}}{P^+}\right)^{v_{H_2}}} \tag{3}$$

where $x_{in}$, and $x_{rp}$ are the respective mole fractions of the reactant and product, respectively; $P_{H_2}$ is the partial pressure of $H_2$ and $\varphi_{H_2}$ is its fugacity coefficient. $K_x$ is the equilibrium constant calculated from the mole fractions of reaction participants.

In our previous work [1], the reactants' activity coefficients in the indole/octahydroindole system were calculated using UNIFAC methods [22]. However, it was found that under the autoclave reaction conditions applied in this work, the activity coefficients of the reactants (see Equation (3)) are not very different at a given temperature; likewise, the temperature dependence of the activity coefficients was not strong. The corresponding $K_\gamma$-products were calculated and used to derive $K_a$ from the *experimental* $K_x$-values [1]. However, due to the similarity of the $\gamma_i$-values, the differences between $K_a$ and the *experimental* $K_x$-values were insignificant. This observation facilitates the treatment of the equilibrium data, and in this work, it was assumed that $K_a = K_x$ and the $K_x$-values were used to calculate the thermodynamic functions of reactions R-I to R-IV.

For reactions R-I to R-IV, the equilibrium concentrations of participating species were measured and their associated $K_x$ constants were derived. The results of the chemical equilibrium study of the reactions given in Figure 2 are summarized in Table 1.

**Table 1.** Reaction parameters and calculated values of the liquid phase reactions R-I to R-IV.

| Reaction [a] | $P_{av}$/atm [b] | $T$/K [c] | $K_a$ [d] | $\pm 2\sigma$ [e] | ln $K_a$ | Feed [f] |
|---|---|---|---|---|---|---|
| R-I | 9.3 | 413.15 | 0.89 | 0.25 | −0.1165 | c |
| | 17.9 | 433.15 | 0.50 | 0.07 | −0.6931 | a |
| | 39.4 | 443.15 | 0.39 | 0.39 | −0.9416 | d |
| | 17.2 | 453.15 | 0.30 | 0.24 | −1.2040 | a, b |
| | 40.6 | 463.15 | 0.24 | 0.61 | −1.4271 | d |
| | 18.5 | 473.15 | 0.19 | 0.04 | −1.6607 | b |
| R-II | 9.5 | 413.15 | 3.24 | 0.22 | 1.1756 | c |
| | 19.2 | 423.15 | 1.50 | 0.02 | 0.4055 | f |
| | 17.7 | 433.15 | 0.47 | 0.04 | −0.7550 | a |
| | 39.3 | 443.15 | 0.33 | 0.01 | −1.1087 | d |
| | 15.7 | 453.15 | 0.23 | 0.02 | −1.4697 | b |
| | 18.4 | 473.15 | 0.08 | 0.01 | −2.5257 | b |
| | 21.2 | 483.15 | 0.03 | 0.002 | −3.5066 | e |
| R-III | 9.0 | 413.15 | 335.5 | 121.6 | 5.8156 | c |
| | 19.5 | 423.15 | 96.35 | 35.2 | 4.5680 | f |
| | 17.8 | 433.15 | 29.31 | 1.38 | 3.3779 | a |
| | 19.5 | 453.15 | 3.18 | 0.72 | 1.1569 | a |
| | 21.5 | 483.15 | 0.16 | 0.05 | −1.8326 | e |
| R-IV | 8.6 | 413.15 | 18.00 | 1.71 | 2.8904 | c |
| | 19.1 | 423.15 | 5.79 | 1.05 | 1.7561 | f |
| | 17.7 | 433.15 | 4.30 | 1.15 | 1.4586 | a |
| | 39.0 | 443.15 | 1.24 | 0.06 | 0.2151 | d |
| | 15.8 | 453.15 | 0.70 | 0.49 | −0.3567 | b |
| | 40.6 | 463.15 | 0.37 | 0.01 | −0.9943 | d |

[a] Reaction species are shown in Figure 2; [b] average system pressure at equilibrium; [c] equilibrium temperature; [d] equilibrium constant; [e] all uncertainties provided in this table are presented as expanded uncertainties (0.95 confidence level, k = 2); [f] feed compositions are presented in Table S1.

An analysis of the dependencies of the equilibrium constants (and their logarithms) for the reaction R-I to R-IV (Figure 4) on temperature corresponds to the enthalpy of the reaction. Equilibrium constants are decreasing with increasing temperature, as can be expected for an exothermal reaction (Figure 4). Reaction R-IV (H4-MI to H8-MI) is thermodynamically comparatively unfavorable within the complete temperature range. In this reaction step, an aromatic, five-membered N-heterocycle (i.e., a pyrrole-like structure) is hydrogenated. This type of reaction is known to have a rather low thermodynamic driving force. Reaction R-II (H0-MI to H4-MI) and R-III (M2-MI to H8-MI) both involve hydrogenation in a homocyclic, aromatic ring. However, as R-III means complete hydrogenation of this ring,

R-II only covers its partial hydrogenation. This is thermodynamically less favorable as the aromatic system is destroyed in both systems, but the last reaction step (the highly favorable hydrogenation of a non-aromatic double bond) is only included in R-III. This fact is reflected in the lower equilibrium constants for R-II compared to R-III.

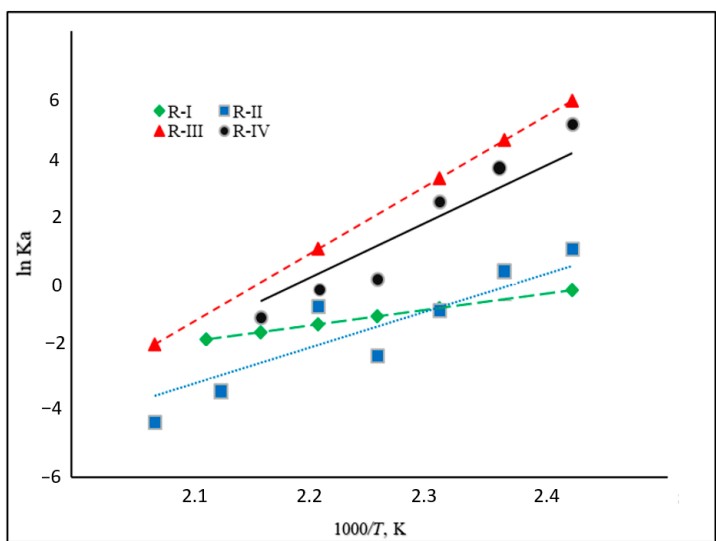

**Figure 4.** Relationships between equilibrium constant and temperature for reactions R-I to R-IV (see Figure 2).

*3.3. Experimental Reaction Thermodynamics in the methyl-indole/octahydro-1-methyl-indole System*

The equilibrium constants of reactions R-I to R-IV relating to the 1-methyl-indole/octahydro-1-methyl-indole system at different temperatures were used to derive the reaction enthalpy (see Table 2) using the Van 't Hoff equation:

$$\frac{d\,lnK_a}{dT} = \frac{\Delta_r H_m^o}{RT^2} \tag{4}$$

$$\ln K_a\,(T) = a \cdot \frac{1000}{T} + b$$
$$\text{with } a = -\frac{\Delta_r H_m^o(T)}{R} \text{ and } b = \frac{\Delta_r S_m^o(T)}{R} \tag{5}$$

where $K_a$ is the liquid phase equilibrium constant; $T$ is the reaction temperature, in K; $\Delta_r H_m^o(T)$ is the standard ($p^\circ = 0.1$ MPa) molar reaction enthalpy, in kJ·mol$^{-1}$; $\Delta_r S_m^o(T)$ is the standard ($p^\circ = 0.1$ MPa) molar reaction entropy change, in J·K$^{-1}$·mol$^{-1}$; $R = 8.31446$ J.K$^{-1}$·mol$^{-1}$ is the gas constant. The thermodynamic properties of reactions R-I to R-IV (i.e., hydrogenation reactions) were calculated using Equation (5) and the results are given in Table 2.

As can be seen from Table 2, the $\Delta_r H_m^o(T)$ values for reactions R-I to R-IV are referenced to the average temperatures, $T_{av}$, of the equilibrium studies, which are in the range of 440–450 K. Already in our previous work [1], it was found for the indole/octahydroindole LOHC system that these reaction enthalpies are indistinguishable (within the *experimental* uncertainties) from the reaction enthalpies calculated from the thermochemical data using Hess's Law and ascribed to the reference temperature $T = 298.15$ K. In addition, for the reaction R-I given in Table 2, the value $\Delta_r H_m^o(443.5 \text{ K}) = -41.7 \pm 1.2$ kJ·mol$^{-1}$ derived from the equilibrium study is very close to the thermochemical value $\Delta_r H_m^o(g, 298.15 \text{ K}) = -45.0 \pm 2.8$ kJ·mol$^{-1}$ calculated with the reliable data on $\Delta_f H_m^o(g, 298.15 \text{ K})$ for 1-methylindole and 1-methylindoline, measured by combustion calorimetry in this work (see Table 3).

**Table 2.** Thermodynamic values of the liquid-phase hydrogenation reactions R-I to R-IV in the 1-methyl-indole/octahydro-1-methy-indole system, as derived from data provided in Table 1 [a].

| Reaction | Method [b] | $T$/K | $\Delta_r H_m^o$/kJ·mol$^{-1}$ | $\Delta_r S_m^o$/J·mol$^{-1}$·K$^{-1}$ |
|---|---|---|---|---|
| R-I | E | 443.5 | $-41.7 \pm 1.2$ | $-101.9 \pm 2.4$ |
| | TC | 298.15 | $-45.0 \pm 2.8$ | |
| | QC | 298.15 | $-43.8 \pm 3.5$ | |
| R-II | E | 442.0 | $-103.4 \pm 6.7$ | $-242 \pm 15$ |
| | QC | 298.15 | $-108.0 \pm 3.5$ | |
| R-III | E | 445.4 | $-181.3 \pm 1.0$ | $-390.5 \pm 2.1$ |
| | QC | 298.15 | $-183.3 \pm 3.5$ | |
| R-IV | E | 450.7 | $-123 \pm 15$ | $-274 \pm 35$ |
| | QC | 298.15 | $-119.1 \pm 3.5$ | |

[a] Reaction species are shown in Figure 2. Uncertainties are expressed as expanded uncertainty (0.95 confidence level, k = 2); [b] method descriptions: E = temperature dependencies of equilibrium constants (Table 1); TC = thermochemistry (i.e., Hess's Law); QC = reaction enthalpy from quantum-chemical calculations.

**Table 3.** Thermochemical data at $T$ = 298.15 K ($p^\circ$ = 0.1 MPa) for indole derivatives (in kJ·mol$^{-1}$).

| Compound | $\Delta_c H_m^o$(liq) [a] | $\Delta_f H_m^o$(liq) | $\Delta_l^g H_m^o$ [b] | $\Delta_f H_m^o$(g)$_{exp}$ | $\Delta_f H_m^o$(g)$_{theor}$ [c] |
|---|---|---|---|---|---|
| 1 | 2 | 3 | 4 | 5 | 6 |
| 1-methyl-indole | $-4921.4 \pm 2.0$ [23] | $93.6 \pm 2.3$ | | | |
| | $-4915.8 \pm 1.4$ [this work] | $87.9 \pm 1.8$ | | | |
| | | $90.1 \pm 1.4$ [d] | $62.0 \pm 0.3$ | $152.1 \pm 1.4$ | $150.7 \pm 2.0$ |
| 1-methyl-indoline | $-5157.7 \pm 1.3$ [this work] | $43.4 \pm 1.8$ | $57.4 \pm 0.5$ | $100.8 \pm 1.9$ | $100.5 \pm 2.0$ |

[a] Uncertainties related to the combustion experiments were estimated according to the procedure recommended in the literature [24–26]; [b] from Table 4; [c] from quantum-chemical calculations; [d] weighted average value (*experimental uncertainty* was taken as the weighting factor).

It is interesting to follow the energetics of the subsequent hydrogenation steps, which are shown in Table 2. A very modest heat effect ($-41.7$ kJ mol$^{-1}$ for the R-I reaction) accompanies the hydrogenation of the double bond in a five-membered N-heterocycle. The hydrogenation of two double bonds in the benzene ring requires approximately the same energy per bond ($-103.4$ kJ mol$^{-1}$ for the R-II reaction, corresponding to $-51.7$ kJ mol($H_2$)$^{-1}$). The hydrogenation of H0-MI to H8-MI by reaction R-III is slightly less exothermic ($-181.3$ kJ mol$^{-1}$, corresponding to $-60.4$ kJ mol($H_2$)$^{-1}$) than hydrogenation of H4-MI by R-IV reaction ($-123$ kJ mol$^{-1}$, corresponding to $-61.5$ kJ mol($H_2$)$^{-1}$). However, the total heat of the reaction for the hydrogenation of H0-MI to H8-MI in the R-I to R-III direction (see Figure 5) is $-223.0$ kJ mol$^{-1}$. Almost the same thermal effect $-226.4$ kJ mol$^{-1}$ (see Figure 5) was obtained for the formation of H8-MI in the direction of R-II to R-IV.

The thermodynamic data obtained in this work and given in Tables 1 and 2 are deemed to be reliable based on the mutual agreement of the values obtained by the different methods.

R-I

R-III

$\Delta_r G_m = +3.5$
$\Delta_r H_m = -41.7 \pm 1.2$
$\Delta_r S_m = -101.9 \pm 2.4$
$T_{av} = 443.5$ K

$\Delta_r G_m = -7.3$
$\Delta_r H_m = -181.3 \pm 1.0$
$\Delta_r S_m = -390.5 \pm 2.1$
$T_{av} = 445.4$ K

$\Sigma \Delta_r H_m = -223.0$
$\Sigma \Delta_r S_m = -492.4$

R-II

R-IV

$\Delta_r G_m = +3.6$
$\Delta_r H_m = -103.4 \pm 6.7$
$\Delta_r S_m = -242 \pm 15$
$T_{av} = 442.0$ K

$\Delta_r G_m = +0.5$
$\Delta_r H_m = -123 \pm 15$
$\Delta_r S_m = -274 \pm 35$
$T_{av} = 450.7$ K

$\Sigma \Delta_r H_m = -226.4$
$\Sigma \Delta_r S_m = -516$

**Figure 5.** Total thermodynamic analysis of conversions in the system 1-methyl-indole/octahydro-1-methylindole. Units of $\Delta_r G_m$ and $\Delta_r H_m$ are kJ·mol$^{-1}$. Units of $\Delta_r S_m$ are J·mol$^{-1}$·K$^{-1}$.

*3.4. Thermodynamic Analysis of Hydrogenation Reactions*

The equilibrium constants (Table 1) and the thermochemical characteristics derived in Table 2 help to perform the thermodynamic analysis of interconversions from the initial substrate over intermediates to the final fully hydrogenated product. The analysis is based on the Gibbs–Helmholtz equation:s

$$\Delta_r G_m^o = -\text{RT} \times \ln K \, (T) = \Delta_r H_m^o \times \Delta_r S_m^o \tag{6}$$

Using the enthalpies and entropies of reactions R-I to R-IV from Table 2, the change of Gibbs energies for these reactions were calculated according to Equation (6) and they are given in Figure 5. These $\Delta_r G_m^o$-values help to understand why the concentrations of both intermediates H2-MI and H4-MI are very low in the equilibrium mixture. Indeed, the slightly positive $\Delta_r G_m^o$-values for reactions R-I and R-II indicate that both these reactions are thermodynamically less favorable than reaction R-III with the negative $\Delta_r G_m^o$-value and reaction R-IV with $\Delta_r G_m^o$-value close to zero as shown in Figure 5. Following, the H2-MI and H4-MI are disappearing from the reaction mixture immediately after their formation and their amounts are limited to equilibrium concentrations.

Another advantage of the thermodynamic analysis is that it helps to assess the degree of $H_2$ release in the individual reactions, which is important for the optimization of the LOHC system. Since the dehydrogenation of perhydro-1-methylindole is a complex system of series-parallel transformations, in order to avoid incorrect interpretation of the results of the analysis, it is necessary to study not only the system as a whole but also each reaction and direction of transformations. In order to quantify the completeness of dehydrogenation, the degree of $H_2$ release (relative to the maximum possible) was determined according to the following methods.

For the individual dehydrogenation reactions in the LOHC system 1-methylindole/octahydro-1-methylindole a thermodynamic analysis was performed for H8-MI in the temperature range of 433–483 K at 1 atm in the hypothetical (solvent-free) gas phase. Separate analyses were performed for each reaction R-I, R-II, R-III, and R-IV (see Figure 6). As an example, if for R-III the quantity H0-MI at equilibrium is defined as *x* moles, then the quantity of H8-MI in that mixture is (1 − *x*) moles and thus the quantity of $H_2$ released is

3*x*. By this logic, the total moles in the mixture is (1 + 3*x*), which leads to Equation (7) from which the equilibrium constant can be calculated:

$$K_x = \frac{x \cdot (3x)^3}{1-x} \cdot \frac{P_{tot}{}^3}{(1+x)^3} = K_a \tag{7}$$

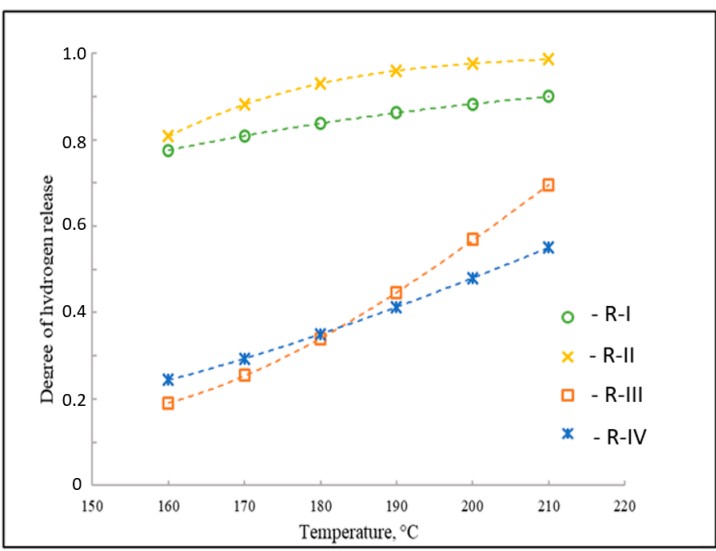

**Figure 6.** Degree of $H_2$ release in the dehydrogenation of individual reactions in the LOHC system 1-methylindole/octahydro-1-methylindole, as referenced to reactions R-I to R-IV given in Figure 2.

By using the value of the *experimental* equilibrium constant (Table 1) for this reaction and solving Equation (7) for *x*, the equilibrium composition is obtained for the dehydrogenation reaction of H8-MI to H2-MI. Similarly, calculations for reactions R-I through R-IV (Figure 2) were performed in this manner. Figure 6 presents the results of the dehydrogenation of this LOHC system.

As a result of the thermodynamic analysis of the dehydrogenation in the system with H0-MI, it was found (Figure 6) that the highest degrees of $H_2$ release at a temperature of 210 °C for the dehydrogenation reactions of H4-MI and H2-MI are equal to about 99% and 90%, respectively. For the two remaining transformations in this system, the degree of $H_2$ release cannot exceed 70% (Figure 6). Therefore, when evaluating the possibility of using an organic molecule as a LOHC, the thermodynamic properties of all reactions taking place in the system must be taken into account, since the thermodynamics of the formation of intermediates make a significant contribution to the composition of the reaction mixture at equilibrium.

### 3.5. The Stability of the Substrates under the Conditions of Hydrogenation–Dehydrogenation

Thermal stability of the reactants in potential LOHC systems is a key criterion as they are supposed to undergo cyclical hydrogenation–dehydrogenation reactions. To understand this behavior in the system of interest, the selectivity was investigated in the 1-methylindole/octahydro-1-methylindole system (*T* = 140–210 °C) and compared against the indole/octahydroindole system (*T* = 150–220 °C). The contact time was sufficient to reach chemical equilibrium in the system. The results are shown in Figure 7. The selectivity was determined by a GC analysis of the reaction mixture and was defined as the ratio between the goal reactants remaining after equilibration and the amount of non-identified decomposition products detected in the GC chart.

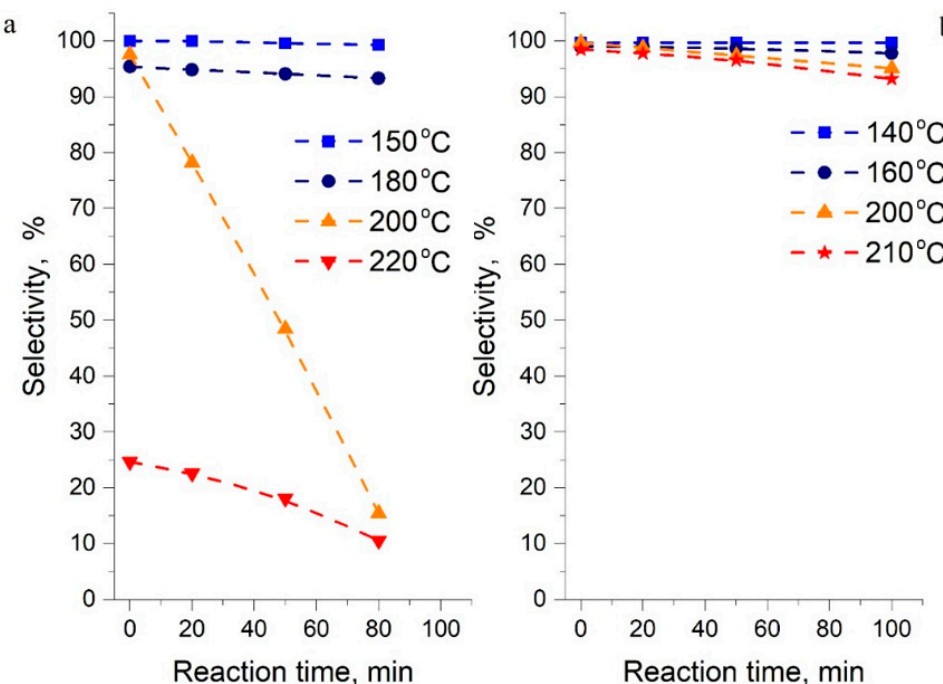

**Figure 7.** Selectivity in the direction of hydrogenation–dehydrogenation: indoline [1] (**a**) and 1-methylindole (**b**). Selectivity for 1-methyl-indole and its non-decomposed (i.e., partially hydrogenated derivatives) in cyclical hydrogenation–dehydrogenation reactions.

It is obvious in Figure 7 that the system based on 1-methylindole is more stable than indoline. For example, at 200 °C for 1-methylindole, the selectivity for hydrogenation reactions decreased from 99.8% to 95% for 100 min of contact, and for indoline from 99.5% to 12% for the same time of contact. As shown in Figure 7, the indole system maintains its stability at temperatures up to a range of 150–180 °C, however, above this range, rapid decomposition of all species in the reaction mixture occurs. Therefore, the maximum operating temperatures of 200 °C and 170–180 °C for 1-methylindole and indole, respectively, are recommended for practical $H_2$ storage with this LOHC system.

### 3.6. Thermochemical Measurements on 1-methyl-indole and 1-methyl-indoline

3.6.1. Combustion Calorimetry

The thermodynamic functions of indole derivatives obtained experimentally from equilibrium studies (see Table 2) can be derived independently from the standard molar enthalpies of formation of the reactants, if these data are available and reliable. We found only one thermochemical study related to the compound of interest in this work. The enthalpy of vaporization and the enthalpy of formation of 1-methylindole were determined by Ribeiro da Silva et al. [23]. In order to validate these results and to provide the missing thermochemical data, the complementary combustion experiments and vapor pressure measurements on 1-methyl-indole and 1-methyl-indoline were carried out in this work.

The standard specific energies of combustion $\Delta_c u^o$(liq) = −37,432.5 ± 3.2 J·g$^{-1}$ for 1-methyl-indole and $\Delta_c u^o$(liq) = −38,677.8 ± 2.3 J·g$^{-1}$ for 1-methyl-indoline were measured from five experiments for each compound. Primary results from combustion experiments are given in the electronic supporting information in Tables S2 and S3. The *experimental* standard molar enthalpies of combustion, $\Delta_c H_m^o$, were used to derive the standard molar enthalpies of formation in the liquid state $\Delta_f H_m^o$(liq), given in Table 3. Values of $\Delta_c H_m^o$ refer to the reactions:

1-methyl-indole: $C_9H_9N$ (cr) + 11.25 $O_2$(g) = 9 $CO_2$ (g) + 4.5 $H_2O$ (liq) + 0.5 $N_2$ (g)  (8)

1-methyl-indoline: $C_9H_{11}N$ (cr) + 11.75 $O_2$(g) = 9 $CO_2$ (g) + 5.5 $H_2O$ (liq) + 0.5 $N_2$ (g)  (9)

Enthalpies of formation $\Delta_f H_m^o$(liq) of 1-methyl-indole and 1-methyl-indoline were calculated from the enthalpic balance according to Equations (8) and (9) using standard molar enthalpies of formation of $H_2O$ (liq) and $CO_2$ (g), recommended by CODATA [24]. For converting the energy of the actual bomb process to that of the isothermal process, and reducing to standard states, the conventional procedure was applied [25]. Uncertainties related to combustion experiments, were calculated according to guidelines presented in [26].

Our results from combustion calorimetry on 1-methyl-indole are in fair agreement (within the combined uncertainties) with those published by Ribeiro da Silva et al. [23] (see Table 3). Thermochemical experiments were carried out with 1-methyl-indoline for the first time.

From our experience, validation of *experimental* thermochemical data is of paramount importance, especially when a "fair" fit of *experimental* data is found or only a single *experimental* result is available. In this century, quantum chemical (QC) calculations are becoming a valuable validation tool for thermochemical data, but since the QC results refer to the gas phase, the enthalpies of vaporization are required to obtain the liquid phase enthalpy of formation, $\Delta_f H_m^o$(liq), which is finally relevant to the energetics of $H_2$ storage. In this work, the high-level QC methods were applied to calculate the gas-phase standard molar enthalpies of formation, $\Delta_f H_m^o$(g, 298.15 K), and determined the vaporization enthalpies, $\Delta_l^g H_m^o$(298.15 K), of 1-methyl-indole and 1-methyl-indoline using the transpiration method, so that the $\Delta_f H_m^o$(liq)-values can be validated with the basic equation:

$$\Delta_f H_m^o(\text{liq, 298.15 K}) = \Delta_f H_m^o(\text{g, 298.15 K}) - \Delta_l^g H_m^o(\text{298.15 K}) \tag{10}$$

The development of both contributions to Equation (10) is shown in the following sections.

### 3.6.2. Vapor Pressure Measurements Using the Transpiration Method

Vapor pressures of 1-methyl-indole and 1-methyl-indoline measured using the transpiration method (see Table S4) were approximated using the following equation [12]:

$$R \times \ln(p_i/p_{ref}) = a + \frac{b}{T} + \Delta_l^g C_{p,m}^o \times \ln\left(\frac{T}{T_0}\right) \tag{11}$$

where $R = 8.31446$ J.K$^{-1}$.mol$^{-1}$ is the universal gas constant, the reference pressure, $p_{ref} = 1$ Pa, and $a$ and $b$ are adjustable parameters; the arbitrary temperature $T_0$ applied in Equation (11) was chosen to be $T_0 = 298.15$ K. $\Delta_l^g C_{p,m}^o$ is the difference of the molar heat capacities of the gas and the liquid phases, respectively (see Table S5). Approximation coefficients of Equation (11) are given in Table S4 and they were used to derive vaporization enthalpies according to this equation:

$$\Delta_l^g H_m^o(\text{298.15 K}) = -b + \Delta_l^g C_{p,m}^o \times 298.15 \tag{12}$$

The combined uncertainties of vaporization enthalpies were calculated as described elsewhere [27,28]. They include uncertainties from the *experimental* conditions of transpiration, uncertainties in vapor pressure, and uncertainties due to the temperature adjustment to $T = 298.15$ K. The compilation of the $\Delta_l^g H_m^o$ -values measured in this work using the transpiration method and those found in the literature is given in Table 4.

As can be seen from Table 4, the $\Delta_l^g H_m^o$(298.15 K)-values for 1-methyl-indole as derived from different techniques agree well and the weighted average value $\Delta_l^g H_m^o$(298.15 K) = 62.0 ± 0.3 kJ·mol$^{-1}$ was recommended for thermochemical calculations.

**Table 4.** Compilation of available enthalpies of vaporization, $\Delta_l^g H_m^o$, for 1-methyl-indole and 1-methyl-indoline.

| Compound/CAS | Method [a] | *T*-Range (K) | $\Delta_l^g H_m^o(T_{av})$ (kJ·mol$^{-1}$) | $\Delta_l^g H_m^o$(298.15 K) [b] (kJ·mol$^{-1}$) | Ref. |
|---|---|---|---|---|---|
| 1-methyl-indole (liq) | DC | 334.3 | 67.1 ± 0.3 | 62.2 ± 1.6 | [23] |
| | CGC | | | 61.1 ± 3.1 | [29] |
| | S | 268.6–341.9 | 61.6 ± 0.1 | 61.9 ± 0.3 | [30] |
| | SC | | | 61.1 ± 2.0 | [31] |
| | T | 294.2–354.2 | 60.7 ± 0.4 | 62.3 ± 0.5 | Table S4 |
| | | | | **62.0 ± 0.3** [c] | average |
| 1-methyl-indoline (liq) | T | 278.4–333.2 | 57.1 ± 0.4 | 57.4 ± 0.5 | Table 1 |

[a] Methods: DC = drop calorimetry; T = transpiration method; CGC = from correlation gas-chromatography; SC = method based on solution calorimetry; S = static method; [b] adjustment to *T* = 298.15 K using heat capacity differences from Table S5; [c] weighted mean value (the uncertainties were taken as the weighing factor).

### 3.6.3. Heat Capacity Measurements

The $\Delta_l^g C_{p,m}^o$-values are involved in the vapor pressure approximation according to Equation (11). These values are usually estimated based on the $C_{p,m}^o$(liq) of *experimental* or empirical origin. As a rule, the liquid-phase heat capacities can be estimated with sufficient accuracy using a group additivity (GA) method [32,33]. However, the limitations of group additivity for cyclic molecules are well known. Hence, it was decided to measure the heat capacity of 1-methylindole to test the reliability of the GA method. Primary *experimental* results for the heat capacities measured in this work for 1-methylindole are given in Table S6 in ESI. The data measured in the 228–350 K range were fitted by the following equation:

$$C_{p,m}^o(\text{liq})/R = 27.07 + 3.139 \times 10^{-2}(T - 298.15/\text{K}) \qquad (13)$$

with $R$ = 8.314462 J·mol$^{-1}$·K$^{-1}$. The *experimental* heat capacity at the reference temperature $T$ = 298.15 K, $C_{p,m}^o$(liq, 298.15 K) = 225.1 J.K$^{-1}$.mol$^{-1}$, agrees very well with the GA estimate $C_{p,m}^o$(liq, 298.15 K) = 228.7 J.K$^{-1}$.mol$^{-1}$ for 1-methyl-indole. Therefore, a value of $C_{p,m}^o$(liq, 298.15 K) = 228.9 J.K$^{-1}$.mol$^{-1}$ was estimated for 1-methyl-indoline. The $\Delta_l^g C_{p,m}^o$-values derived from the heat capacity results are given in Table S5 and they have been used for the temperature adjustments of *experimental* vaporization enthalpies of 1-methyl-indole and 1-methyl-indoline (see Table 4).

### 3.6.4. *Theoretical* Gas-Phase Formation Enthalpies of Intermediates in the 1-methyl-indole/octahydro-1-methyl-indole LOHC System

The *experimental* gas-phase enthalpies of formation of 1-methyl-indole and 1-methyl-indoline were derived from the combustion calorimetry results on $\Delta_f H_m^o$(liq, 298.15 K) and transpiration method results on $\Delta_l^g H_m^o$(298.15 K). These values are given in Table 3, column 5 and they can now be compared with the results from quantum chemical calculations. The high-level methods G4, G4MP2, G3MP2, and CBS-APNO methods were used to get the *theoretical* $\Delta_f H_m^o$(g, 298.15 K)-values of reactions R-I to R-IV participants. The UFF method was used to find the most stable conformers of indole derivatives (see Table 5) [34], which were then optimized by using G3MP2, with enthalpies $H_{298}$ for each were calculated by using and G4, G4MP2, G3MP2, and CBS-APNO methods.

The $H_{298}$-values have been converted to the standard molar enthalpies of formation $\Delta_f H_m^o$(g, 298.15 K)$_{theor}$ using the atomization (AT), as well as using the "well-balanced reactions" (*WBR*) [35]. For the *WBR* method, three reactions were designed for each indole derivative (they are given in Tables S7–S11 ESI). Using reliable *experimental* $\Delta_f H_m^o$(g, 298.15 K)$_{exp}$-values of the reference compounds (see Table S12), the *theoretical* gas-phase enthalpies were calculated using Hess´s Law. The results of QC calculations are summarized in Table 6.

**Table 5.** Most stable conformers (G4 calculations) of reaction intermediates of R-I to R-IV and their gas phase formation enthalpies.

| Indole | Structure | $\Delta_f H_m^o(g)_{G4}$/ [a] kJ·mol$^{-1}$ |
|---|---|---|
| 1-methylindole CAS 603-76-9 | | 149.9 |
| 1-methylindoline CAS 88475-55-2S | | 106.1 |
| 1-methyl-4,5,6,7-tetrahydro-1H-indole | | 41.9 |
| trans-1-methyl-octahydroindole | | −77.2 |
| cis-1-methyl-octahydroindole | | −74.8 |

[a] Calculated using the atomization procedure.

As can be seen from Table 6, the enthalpies of formation calculated using G4, G4MP2, G3MP2, and CBS-APNO are very close, regardless of whether an atomization or a WBR method is applied. The quantum-chemical values for each compound were averaged and the *theoretical* values $\Delta_f H_m^o(g, 298.15 \text{ K})_{\text{theor}}$ (see Table 6, last column) were obtained. These *theoretical* values are compared with the *experimental* values $\Delta_f H_m^o(g, 298.15 \text{ K})_{\text{exp}}$ in Table 4 and the agreement is within the limits of *experimental* uncertainties.

H0-MI, H2-MI, H4-MI, and H8-MI are involved in the reactions R-I to R-IV. Thus, the *theoretical* reaction enthalpies were calculated (see Table 2) using the *theoretical* enthalpies of the formation of reaction participants according to Hess´s Law. The comparison of these "*theoretical*" reaction enthalpies ("QC" in Table 2) with those from chemical equilibrium

experiments ("E" in Table 2) and the results from thermochemistry ("TC" in Table 2) confirms good consistency of results within the *experimental* errors. Such agreement is a necessary indicator by which to validate the assumptions used to treat chemical equilibrium results presented in Section 3.2.

**Table 6.** *Theoretical* gas-phase enthalpies of formation $\Delta_f H_m^o(g)$ at $T = 298.15$ K ($p^\circ = 0.1$ MPa) for indole derivatives as calculated by different methods (in kJ·mol$^{-1}$).

| Compound | G4 AT [a] | G4 WBR [b] | G4MP2 AT [a] | G4MP2 WBR [b] | G3MP2 AT [a] | G3MP2 WBR [b] | CBS-APNO WBR [b] | $\Delta_f H_m^o(g)_{theor}$ [c] |
|---|---|---|---|---|---|---|---|---|
| N-methyl indole | 149.9 ± 3.5 | 151.1 ± 2.0 | 146.7 ± 3.5 | 152.7 ± 2.0 | 148.9 ± 4.1 | 152.0 ± 2.4 | 149.1 ± 2.4 | 150.7 ± 2.0 |
| N-methyl indoline | 106.1 ± 3.5 | 98.6 ± 2.0 | 104.3 ± 3.5 | 100.1 ± 2.0 | 104.3 ± 4.1 | 99.2 ± 2.4 | 99.2 ± 2.4 | 100.5 ± 2.0 |
| H4-N-methylindole | 41.9 ± 3.5 | 36.6 ± 2.5 | 41.4 ± 3.5 | 37.7 ± 2.5 | 43.9 ± 4.1 | 39.4 ± 2.9 | 41.0 ± 2.9 | 39.5 ± 2.2 |
| cis-H8-N-methylindole | −74.8 ± 3.5 | −83.5 ± 1.8 | −73.6 ± 3.5 | −82.3 ± 1.8 | −72.2 ± 4.1 | −82.6 ± 2.1 | −84.2 ± 2.1 | −81.5 ± 1.8 |
| trans-H8-N-methylindole | −77.2 ± 3.5 | −85.9 ± 1.8 | −75.9 ± 3.5 | −84.6 ± 1.8 | −73.7 ± 4.1 | −84.1 ± 2.1 | −82.1 ± 2.1 | −82.9 ± 1.8 |

[a] Calculated by the G4, G4MP2, or G3MP2 methods according to the standard atomization procedure. The expanded uncertainty assessed to be ±3.5 kJ·mol$^{-1}$ for the G4 [13]. The expanded uncertainty assessed to be ±4.1 kJ·mol$^{-1}$ for the G3MP2 [15] and CBS-APNO methods; [b] calculated by the G4, G4MP2, G3MP2, and CBS-APNO methods with help of well balance reactions (see ESI) using the *experimental* $\Delta_f H_m^o(g)$-values for the reaction participants (see Tables S7–S11); [c] calculated for each compound as the weighted average from columns 2 to 8 from this table.

### 3.6.5. Thermodynamic Analysis of the Dehydrogenation/Hydrogenation of the LOHC Systems

As a last step, both contributors ($\Delta_f H_m^o(g, 298.15$ K) and $\Delta_l^g H_m^o(298.15$ K)) to Equation (10) were validated. Thus, reliable liquid-phase enthalpies of formation for the participants of hydrogenation/dehydrogenation reaction of 1-methyl-indole (see Figure 8) can be provided now, as well as reliable hydrogenation/dehydrogenation enthalpy for this reaction in the liquid phase.

**Figure 8.** Hydrogenation/dehydrogenation reaction of 1-methyl-indole in the liquid phase.

Thermochemical data required for these calculations are given in Table 7 for the hydrogen-rich (HR) species.

**Table 7.** Liquid phase enthalpies of formation, $\Delta_f H_m^o(liq)$, of the hydrogen-reach counter-parts, at $T = 298.15$ K ($p^\circ = 0.1$ MPa, in kJ·mol$^{-1}$).

| Compound | $\Delta_f H_m^o(gas)_{HR}$ | $\Delta_l^g H_m^o$ | $\Delta_f H_m^o(liq)_{HR}$ [a] |
|---|---|---|---|
| H8-indole [36] | −64.0 ± 1.9 | 53.5 ± 0.7 | −117.5 ± 2.0 |
| H8-MI | −82.9 ± 1.8 [b] | 49.2 ± 0.8 [c] | −132.1 ± 2.0 |

[a] Calculated according to Equation (10); [b] from Table 6; [c] estimated as shown in Figure S4.

Thermochemical data required for these calculations are given in Table 8 for the for their hydrogen-lean (HL) counter-parts.

The hydrogenation enthalpy of the H0-1-methyl-indole/H8-1-methyl-indole LOHC system (see Figure 8) was derived according to Hess's Law, using the standard molar enthalpies of formation of the reaction participants evaluated in Tables 7 and 8 and listed in Table 9. The hydrogenation enthalpy of the H0-indole/H8-indole LOHC system is given in Table 9 for comparison.

**Table 8.** Liquid phase enthalpies of formation, $\Delta_f H_m^o$(liq), of the hydrogen-lean counter-parts, at $T = 298.15$ K ($p^\circ = 0.1$ MPa, in kJ·mol$^{-1}$).

| Compound | $\Delta_f H_m^o$(gas)$_{HL}$ | $\Delta_l^g H_m^o$ | $\Delta_f H_m^o$(liq)$_{HL}$ [a] |
|---|---|---|---|
| indole [36] | $162.5 \pm 1.0$ | $65.3 \pm 0.6$ | $97.2 \pm 1.2$ |
| H0-MI [Table 4] | $152.1 \pm 1.4$ | $62.0 \pm 0.3$ | $90.1 \pm 1.4$ |

[a] Calculated according to Equation (10).

**Table 9.** Liquid phase reaction enthalpies, $\Delta_r H_m^o$(liq), of the hydrogenation of perhydro-indoles (reaction according to Equation (8)), at $T = 298.15$ K ($p^\circ = 0.1$ MPa, in kJ·mol$^{-1}$).

| Compound | $\Delta_f H_m^o$(liq)$_{HR}$ [a] | $\Delta_f H_m^o$(liq)$_{HL}$ [b] | $\Delta_r H_m^o$(liq) [c] | $\Delta_r H_m^o$(liq)/H$_2$ [d] |
|---|---|---|---|---|
| H8-indole [36] | $-117.5 \pm 2.0$ | $97.2 \pm 1.2$ | $-214.7 \pm 2.3$ | $-53.7 \pm 0.8$ |
| H8-MI | $-132.1 \pm 2.0$ | $90.1 \pm 1.4$ | $-222.2 \pm 2.4$ | $-55.6 \pm 0.8$ |

[a] From Table 7; [b] from Table 8; [c] calculated according to the Hess´s Law applied to Figure 8; [d] reaction enthalpy per mole H$_2$.

From a practical point of view, relating the enthalpy of the reaction to the amount of $H_2$ released (kJ·mol$^{-1}$/H$_2$) allows a comparison of the enthalpy values of LOHC systems with different stoichiometries. In these units, the hydrogenation enthalpies of indole and 1-methyl-indole $-53.7$ and $-55.6$ kJ·mol$^{-1}$/H$_2$ (see Table 9) are quite comparable within their uncertainties. However, the thermal stability of N-methyl-indoles is significantly higher, giving more advantages for technical applications of the latter series of LOHC.

## 4. Conclusions

In this work, a thermodynamic study was conducted on 1-methyl-indole as a LOHC compound. Reaction thermodynamics have been analyzed by two approaches: evaluations of reaction equilibria and combustion calorimetry, validated by quantum-chemical methods.

The heat demand due to the enthalpy of reaction for $H_2$ release from perhydro-1-methyl-indole is significantly lower than in the case of homocyclic LOHCs and in a similar order of magnitude than for perhydro-indole. Hence, it can be concluded that the dehydrogenation reaction can be performed under similarly mild conditions as for perhydro-indole. Furthermore, the results support the hypothesis that methylated indole is more stable in the reactive cycle of hydrogenation and dehydrogenation. The amount of decomposition products observed for the 1-methyl-indole LOHC system is significantly lower than for the indole system under the same conditions. The methylation of the carrier molecule improves the chemical stability in the reactive cycle. This observation might also be transferable to other LOHC systems. However, a small disadvantage of the methylated carrier is the slightly higher molar mass, corresponding to a marginally smaller gravimetric storage density.

**Supplementary Materials:** The following supporting information can be downloaded at: https: //www.mdpi.com/article/10.3390/appliedchem3010004/s1. Table S1: The composition of the initial mixtures for the study of chemical equilibrium; Table S2: Formula, density $\rho$($T$ = 293 K), and massic heat capacity c$_p$ ($T$ = 298.15 K) and expansion coefficients ($\delta V/\delta T$)$_p$ of the materials used in the present study; Table S3: Results for typical combustion experiments at $T$ = 298.15 K ($p$ = 0.1 MPa); Table S4: Results of transpiration method for indole derivatives: absolute vapor pressures $p$, standard molar vaporization enthalpies and standard molar vaporization entropies; Table S5: Compilation of data on molar heat capacities $C^o_{p,m}$(liq) and heat capacity differences; Table S6: Experimental molar heat capacities $C^o_{p,m}$(liq) (in J.K$^{-1}$.mol$^{-1}$) of 1-methylindole; Table S7: Reactions and reaction enthalpies calculated by using quantum-chemical methods for 1-methyl-indole; Table S8: Reactions and reaction enthalpies calculated by using quantum-chemical methods for 1-methyl-indoline; Table S9: Reactions and reaction enthalpies calculated by using quantum-chemical methods for H4-1-methyl-indole; Table S10: Reactions and reaction enthalpies calculated by using quantum-chemical methods for cis-H8-1-methyl-indole; Table S11: Reactions and reaction enthalpies calculated by using quantum-chemical methods for trans-H8-1-methyl-indole; Table S12: Reference values for $\Delta_f H^o_m$(g, 298.15 K) used for calculation reaction enthalpies in Tables S7–S11 with help of quantum-chemical methods. Figure S1: Reaction scheme employed to produce 1-methylindoline; Figure S2: NMR spectra of 1-methylindoline; Figure S3: Comparison of vapor pressures for1-methyl-indole; Figure S4: Calculation enthalpy of vaporization, $\Delta_l^g H^o_m$(298.15 K), of H8-1-methyl-indole based on the vaporization enthalpy of H8-indole. References [37–46] are cited in the supplementary materials.

**Author Contributions:** S.V.V.: conceptualization, methodology, investigation, formal analysis, writing—review and editing. M.E.K.: investigation, writing—review and editing. V.V.T.: formal analysis, calculations. K.M.: writing—original draft, formal analysis, methodology. J.E.B.: synthesis, writing—review and editing. S.P.V.: writing—original draft, formal analysis, methodology, project administration. All authors have read and agreed to the published version of the manuscript.

**Funding:** This research was funded by German Science Foundation (DFG) in the frame of the priority program SPP 1807 "*Control of London Dispersion Interactions in Molecular Chemistry*", grant VE 265-9/2 for S.P.V. The work was supported by the Ministry of Science and Higher Education of the Russian Federation (theme No. AAAA-A12-1111100072-9) as part of the state task of the Samara State Technical University (creation of new youth laboratories). This paper has been supported by the Kazan Federal University Strategic Academic Leadership Program ("PRIORITY-2030").

**Institutional Review Board Statement:** Not applicable.

**Informed Consent Statement:** Not applicable.

**Data Availability Statement:** All data used in this paper are given in the main text and in the Supplementary Materials.

**Conflicts of Interest:** The authors declare no conflict of interest.

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
