# Peer review of "Thermodynamics of Hydrogen Storage: Equilibrium Study of Liquid Organic Hydrogen Carrier System 1-Methylindole/octahydro-1-methylindole"

_appliedchem, doi:10.3390/appliedchem3010004_

Round 1

Reviewer 1 Report

This manuscript has clearly stated goal and scope, properly described experimental methodology and adequately deep analysis of data. Although it contains many quantitative information on some places it is still quite readable and can be of value for the interested readership. Before the final acceptance I have only a few minor comments which should improve the original text:

Eq. (4) and (5) and around – in fact. standard enthalpy and entropy are present; some comments on the (selected, related) standard state(s) would be desirable.

Caption to Fig. 5 should contain units of the displayed thermodynamic quantities.

Owing to Introduction section I would expect in Conclusions some comments of the relevance of findings of this paper for the LOHC-based technologies.

Reviewer 2 Report

This is an excellent experimental and theoretical equilibrium study of LOHC system 1-methylindole/octahydro-1-methylindole. The article is the result of the joint work of well-known experts in various fields of chemical thermodynamics. The results obtained are mutually consistent. The discussion is interesting. The uncertainty values and the number of significant digits are given correctly. The article can be accepted for publication after some minor corrections.

1) Page 2. At the end of section 2.1, a reference should be given to Table S2 (results of determining the water mass fraction).

2) Page 8. In equations (4) and (5), the size of the symbols should be corrected and the constant K should be replaced with Ka.

3) Page 9. In the title of Figure 5, the dimensions of the thermodynamic parameters presented (Gibbs energy, enthalpy and entropy of reactions) should be indicated.

4) Page 9. In the sentence after equation (6), it is necessary to replace the Gibbs enthalpies with Gibbs energies. In the same paragraph, the authors write that negative Gibbs energy values are observed for reactions R-III and R-IV, but for the reaction R-IV, Figure 5 shows a positive value of this parameter.

5) Page 10. There are no line designations in Figure 6 (as in Figure 4).

6) Page 11. Figure 7a is made for indoline, and Figure 7b for 1-methylindole (and not vice versa as in the name of the figure).

7) Page 16. Table 9. In the fourth column, the values should be given with a minus sign. In the fifth column, the uncertainties of the values should be given.

Reviewer 3 Report

Title: Thermodynamics of hydrogen storage: equilibrium study of LOHC system 1-methylindole/octahydro-1-methylindole

Description: The authors investigated an important topic of hydrogen storage developments. They mentioned that the methylated indoles are promising candidates and they could be used for hydrogen storage based on the Liquid Organic Hydrogen Carrier approach. Accordingly, they investigated whether these expectations are justified by analyzing the chemical equilibrium of hydrogenation/dehydrogenation reactions in the indole/octahydroindole LOHC system in a wide range of temperature, pressure, and hydrogen:feedstock ratio. They found thermodynamic properties of chemical reactions and the standard molar enthalpies of formation of the species involved in the reaction have been determined experimentally utilizing combustion calorimetry and additionally validated using high-level quantum chemical methods. The evaluation confirms both hypothesis: 1.) 1-methyl indole exhibits less decomposition during reaction as is the case for the indole system. 2.) The enthalpy of reaction for H2 release from octahydro 1 methyl indole is estimated to be about +55.6 kJ mol(H2) 1 for reaction in the liquid phase at standard conditions. This is comparable to the values observed for octahydroindole.

Comment: This is an important work regarding the issue of hydrogen storage developments, but I can advise the authors to revise their work.

1- The title is not informative enough to attract the readers, please modify the title by adding some extra words to show benefits of performing your investigation. I doubt if most of readers know LOHC as mentioned in the title.

2- Abstract needs a serious modification. Please eliminate general texts from the current abstract, please add your main goal of this study at the beginning part of abstract. Please mention about employed procedures and the highlighted achievements. I mean please emphasize more on the details of current work in the abstract.

3- In the Introduction part, you can make some explanation for employing computational tools as a complementary of experimental studies and you may insert some references for the interested readers, for example you may use this reference:    https://doi.org/10.1016/j.molstruc.2016.06.053    ,   or so many other examples.

4- You may include the XYZ coordinates of the optimized models in the supplementary file.

5- Can you give us some explanations on possible environmental effects on the results? I mean, you want to adsorb the H2 substance, but what about other substances in the mixture of gases, how can you make a separation? Or, any other effects such as humidity are important or no?

6- You may also make a judgment based on the stability of indole derivatives (boiling point, …) and their existence in different environment.

7- Is there any possibility for enhancing other catalytic surfaces by loading these indole derivatives?

8- Adsorption/Desorption? Can you make some explanation about the recovery time of the derivatives? Are they used as permanent storage or temporarily or programmable?

9- Good luck! 
